# Decade-Long Sustained Cellular Immunity Induced by Sequential and Repeated Vaccination with Four Heterologous HIV Vaccines in Rhesus Macaques

**DOI:** 10.3390/vaccines13040338

**Published:** 2025-03-21

**Authors:** Xiaozhou He, Danying Chen, Qi Ma, Yanzhe Hao, Hongxia Li, Xiaoguang Zhang, Yuxi Cao, Xia Feng

**Affiliations:** 1National Key Laboratory of Intelligent Tracking and Forecasting for Infectious Disease, National Institute for Viral Disease Control and Prevention, Chinese Center for Disease Control and Prevention, Beijing 102206, China; hexz@ivdc.chinacdc.cn (X.H.); chendanying@ccmu.edu.cn (D.C.); zhangxg@ivdc.chinacdc.cn (X.Z.); 2Beijing Key Laboratory of Viral Infectious Disease, Institute of Infectious Diseases, Beijing Ditan Hospital, Capital Medical University, Beijing 100015, China

**Keywords:** HIV vaccine, heterologous prime–boost, cellular immunity, rhesus macaques, longitudinal study

## Abstract

Background/Objectives: Developing durable cellular immunity remains a critical challenge for HIV vaccine development. Methods: We evaluated a sequential and repeated heterologous prime–boost vaccination regimen using four distinct vector-based vaccines (DNA, rAd5, rSeV, and rMVA) expressing HIV-1 gag in rhesus macaques over a decade-long observation period. Results: Compared to the two-vector and control groups, the four-vector regimen elicited potent gag-specific cellular immune responses, as evidenced by IFN-γ ELISPOT assays showing sustained responses exceeding 500 SFCs/10^6^ PBMCs for up to 52 or 69 weeks post-vaccination. Intracellular cytokine staining revealed multifunctional CD4+ and CD8+ T-cell responses, while humoral immunity against Ad5 vectors remained manageable despite repeated administrations. Conclusions: These findings demonstrate that sequential and repeated heterologous vaccination effectively induces and maintains durable cellular immunity, providing a strategic framework for HIV vaccine design.

## 1. Introduction

Acquired immunodeficiency syndrome (AIDS), caused by the human immunodeficiency virus (HIV), remains a major global health challenge. Despite extensive international efforts, an estimated 39.9 million people were living with HIV in 2023, with 1.3 million new infections reported in that year alone. In the absence of a cure, developing safe and effective vaccines remains crucial for controlling the persistent HIV/AIDS epidemic [1]. Over the years, the global response to infectious diseases, particularly HIV/AIDS, has driven the development of innovative vaccine strategies [2]. Five vaccine concepts have been tested over the course of 40 years in nine clinical efficacy trials, none of which showed high efficacy [3]. Bivalent HIV-1 Env Gp120 proteins AIDSVAX B/E and AIDSVAX B/B were tested in early efficacy trials VAX003 and VAX004 and failed to show efficacy [4,5,6]. The Ad5 vector expressing HIV-1 Gag, Pol, and Nef was unsuccessful in conferring protection against HIV-1 infection in the efficacy trials HVTN 502 and HVTN 503, which were focused on cell-mediated immunity [7,8,9]. A pox-protein prime–boost strategy was used in the efficacy trial RV144. While the trial did not meet prespecified efficacy endpoints, a post hoc modified intention-to-treat analysis revealed a 31.2% efficacy rate in secondary analyses [1]. Subsequent investigations, such as the HVTN 505 trial, evaluated a multiclade DNA prime followed by a recombinant Ad5 boost targeting multiple HIV antigens. Despite inducing antigen-specific immune responses, the regimen failed to correlate with protective outcomes, leading to early termination [10]. Similarly, the HVTN 702 in South Africa—a direct successor to RV144—also reported no protective efficacy, underscoring challenges in extrapolating RV144 findings from Thailand to distinct epidemiological settings [11,12]. The tetravalent Ad26 vector expressing HIV mosaic immunogen prime with a protein boost was tested in the most recent efficacy trials HVTN 705 (Imbokodo) and HVTN 706 (Mosaico). These vaccine regimens did not provide significant protection against HIV-1 infection [13,14]. However, maintaining durable immune protection remains a significant challenge, and the long-term efficacy of single-dose vaccines remains uncertain [15,16]. These limitations underscore the urgent need for novel vaccination approaches capable of inducing sustained immune responses [17,18].

The heterologous prime–boost immunization strategy has emerged as a promising approach in recent years [19]. Originally developed for HIV vaccine research, this strategy has now been successfully applied to combat other diseases such as malaria and tuberculosis, and has shown potential in cancer immunotherapy [20,21]. The approach involves the sequential administration of different vaccine platforms to enhance immune responses. Pioneering studies in both HIV vaccine development and cancer immunotherapy have demonstrated that this sequential vaccination strategy elicits more robust and durable immune responses compared to single-platform vaccinations. Specifically, the combining of diverse vaccine platforms, such as DNA, viral vectors, and protein-based vaccines, has been shown to induce broader and more potent immune responses than homologous vaccination regimens [22,23]. In the field of cancer immunotherapy, heterologous prime–boost regimens have demonstrated enhanced antitumor immunity and improved clinical outcomes in patients [22,23].

Rhesus macaques, despite limitations in modeling HIV-specific tropism, provide a critical platform for evaluating vaccine durability, as their immune systems closely mirror humans in cellular and humoral response dynamics. In this study, we systematically assessed a sequential heterologous regimen using four vector-based vaccines (DNA, rAd5, rSeV, rMVA) expressing HIV-1 gag over a decade-long observation period. Our findings reveal sustained gag-specific T-cell responses and controlled vector-specific immunity, advancing insights into the interplay between repeated heterologous boosting and long-term immune persistence. This work not only informs HIV vaccine design but also contributes to broader applications in combating rapidly evolving pathogens and enhancing cancer immunotherapy strategies.

## 2. Materials and Methods

### 2.1. Development and Composition of the Four HIV Vaccines

The HIV-1 subtype B’ gag gene, derived from prevalent strains in Henan Province, China, was codon-optimized for mammalian expression to enhance protein production. Based on this optimized B’ gag gene, three types of vaccines were constructed: plasmid DNA, recombined Adenovirus type 5 (rAd5), and recombined Sendai virus (rSeV) vaccines. Firstly, the B’ gag gene was cloned into the pVR plasmid and rAd5 backbone to produce the DNA and rAd5 vaccines, respectively. These vaccines were manufactured by Benyuan Zhengyang Gene Technology Co., Ltd. (Beijing, China) and Shenzhen Yuanxing Gene Technology Co., Ltd. (Shenzhen, China). The SeV/F vector, capable of expression both wild-type and codon-modified gag proteins to induce immune responses, was used to construct the rSeV vaccine. This recombination Sendai virus was developed by DNAVEC Corp. (Tsukuba, Japan). Notably, the amino acid sequences of the wild-type and the optimized B’ gag showed 92% identity.

To further enhance HIV immunogenicity, the B’/C recombinant subtype gag-pol and env genes were cloned from a dominant strain isolated from high HIV prevalence areas in Guangxi province, China. Potential inhibitory sequences were removed through gene optimization to maximize antigen expression. The modified gag-pol and env genes were subsequently synthesized and inserted into recombinant Vaccinia virus vectors, yielding the rMVA vaccine, were produced by Changchun BCHT Pharmaceutical Co., Ltd. (Changchun, China).

### 2.2. In Vitro Assessment of Gag Immunogen in Four Different Vaccines

To assess HIV-1 gag protein expression, HEK293T cells were infected with rAd5, rSeV, or rMVA vaccines at a multiplicity of infection (MOI) of 10. The HEK293T cells were preserved in the Academician Zeng Yi’s laboratory at the National Institute for Viral Disease Control and Prevention, Chinese Center for Disease Control and Prevention. For the DNA vaccine, cells were transfected using FuGene HD transfection regent (Roche, Basel, Switzerland) according to the manufacturer’s instructions. Forty-eight hours post-transfection or infection, cells were harvested, and lysates were prepared for protein analysis. Proteins were separated using sodium dodecyl sulfate polyacrylamide gel electrophoresis (SDS-PAGE) and subsequently transferred onto polyvinylidene difluoride (PVDF, Merck, Darmstadt, Germany) membranes using standard electroblotting procedures.

For gag protein detection, membranes were first blocked with 5% fat-free milk in phosphate-buffered saline containing 0.05% Tween 20 (PBST). The membranes were stripped and sequentially probed with mouse anti-HIV p24 monoclonal antibodies (US National Institutes of Health [NIH] AIDS Research and Reference Reagent Program, Germantown, MD, USA) and mouse anti-GAPDH monoclonal antibodies (Zhongshan Goldenbridge Biotechnology Co., Ltd., Beijing, China). After washing, the membranes were incubated with HRP-conjugated goat anti-mouse IgG antibodies, and proteins were visualized by staining with 3,3′–diaminobenzidine (DAB, Zhongshan Golden Bridge, Beijing, China). GAPDH was used as an internal control to normalize protein quantities.

### 2.3. Rhesus Macaque Groups and Sequential Immunization Schedule

The prime–boost–boost immunization strategy employed for the rhesus macaques in this study has been previously described [24]. Eight adult female rhesus macaques were allocated into three experimental groups (Figure 1), macaque IDs and ages at first vaccination: 98R0053 (7 years), 00R0019 (5 years), 98R0013 (7 years), 00R0227 (5 years), 99R0035 (6 years), 98R0037 (7 years), 98R0009 (7 years) and 00R0023 (5 years).

The four-vector vaccine group (*n* = 5, IDs: 98R0053, 00R0019, 98R0013, 00R0227 and 99R0035, aged 5–7 years) received two doses of each vaccine according to the following schedule: (1) DNA vaccine administration at weeks 0 and 2; (2) rAd5 (or rSeV) vaccine at weeks 6 and 33; (3) rSeV (or rAd5) vaccine at weeks 45 and 87; and (4) rMVA vaccine at weeks 206 and 236. Additionally, these macaques received a second round of rAd5 vaccinations at weeks 282 and 336.

The two-vector vaccine group (*n* = 1, ID: 98R0037, aged 7 years) received intramuscular injections of 1 mL sterile PBS at weeks 0, 2, 6, 33, 45, and 87 as a control for the initial immunization phase. Subsequently, this macaque was immunized with the rMVA vaccine at weeks 206 and 236, followed by rAd5 vaccination at weeks 282 and 336.

The PBS control group (*n* = 2, IDs: 98R0009 and 00R0023, aged 5–7 years) received 1 mL intramuscular injections of sterile PBS at all immunization time points: weeks 0, 2, 6, 33, 45, 87, 206, 236, 282, and 336.

The vaccines were administered at the following doses: DNA, 5 mg/monkey; rAd5, 10^9^ PFU/monkey; rSeV, 10^9^ CIU/monkey and rMVA, 9 × 10^7^ PFU/monkey. All animal procedures were approved by the Ethics Committee of the National Institute for Viral Disease Control and Prevention at the Chinese Center for Disease Control and Prevention (approval number: 20190814029, approval Date: 14 August 2019). Moreover, all experiments were conducted in accordance with the guidelines set forth in the U.K. Animals (Scientific Procedures) Act of 1986 and in the EU Directive 2010/63/EU for animal experiments.

### 2.4. Assessment of Gag-Specific Cellular Immune Responses in Immunized Rhesus Macaques Using ELISPOT Assay

Cellular immune responses were assessed using the enzyme-linked immunospot (ELISPOT) assay. Peripheral venous blood (10 mL) was collected from each rhesus macaque at designated time points post-immunization and anticoajulated with ethylene diamine tetraacetic acid (EDTA). Peripheral blood mononuclear cells (PBMCs) were isolated within 8 h of collection using a commercial monkey lymphocyte separation solution (EZ-Sep Monkey 9X, Shenzhen Dakewe Biotech Co., Ltd., Shenzhen, China). IFN-γ-secreting lymphocytes were quantified following stimulation with a gag-specific peptide pool.

The gag269 peptide pool (University of Tokyo, Tokyo, Japan) consisted of 124 overlapping 14–16 amino acid peptides with an 11-residue overlap between adjacent peptides. Peptide purity ranged from 52% to 100%, and stock solutions were prepared in dimethyl sulfoxide (DMSO). The pool was divided into five sub-pools based on their position relative to the N- and C-terminals: AB (aal-105), CD (aa95-203), EF (aa193-305), GH (aa295-408), and IJ (aa398-516). Each sub-pool was diluted to a final concentration of 2 µM before addition to 96-well ELISPOT plates pre-coated with IFN-γ capture antibodies.

For the assay, 50 μL of each sub-pool solution was added to designated wells, along with negative controls (1% DMSO) and positive controls (50 ng/mL phorbol 12-myristate 13-acetate [PMA] and 1 µg/mL ionomycin). Subsequently, 50 µL of PBMC suspension (4 × 10^6^/mL cells/mL) was added to each well. Plates were covered and incubated at 37 °C with 5% CO_2_ for 36 h. Following incubation, biotin-labeled primary antibodies and HRP-labeled streptavidin were sequentially added. Spots were developed using aminoethyl carbazole (AEC) substrate, and the plates were air-dried before analysis with an ImmunoSpot Reader (CTL, Cleveland, OH, USA). A positive response was defined as spot counts exceeding four times the negative control value (>50 spot-forming cells [SFCs]/10^6^ PBMCs).

### 2.5. Evaluation of Gag-Specific Cellular Immune Responses in Immunized Rhesus Macaques Using Intracellular Cytokine Staining (ICS)

Gag-specific cellular immune responses were analyzed by intracellular cytokine staining (ICS) using frozen PBMC samples collected at weeks 248 and 284 post-primary immunization. Cells were thawed in complete Roswell Park Memorial Institute (RPMI)-1640 medium supplemented with 10% FBS and cultured overnight at 37 °C. For stimulation, 200 μL aliquots of cell suspension (1 × 10^6^ cells) were transferred to 96-well U-bottom plates with the following reagents: 1.25 μg/mL of anti-CD28 (clone 340975, BD Biosciences, Franklin Lakes, NJ, USA) and 1.25 μg/mL anti-CD49d (clone 340976, BD Biosciences, Franklin Lakes, NJ, USA). The gag269 peptide pool was added at a final concentration of 2 μM per peptide, while mock controls received an equivalent volume of DMSO.

Following a 2 h incubation, monensin and brefeldin A (both from Sigma-Aldrich, St. Louis, MO, USA) were added to final concentrations of 2 μM and 5 μg/mL, respectively. After an additional 5 h incubation, 20 μL of 20 mM EDTA was added to each well, and cells were washed with PBS containing 1% FBS.

For surface staining, cells were incubated for 30 min with appropriate volumes of fluorescein isothiocyanate (FITC) mouse anti-human CD3ε (BD biosciences), allophycocyanin (APC)-eFluor 780 anti-human CD8 (eBioscience, Thermo Fisher Scientific, Waltham, MA, USA), and perCP-Cy5.5 anti-human CD4 antibodies (eBioscience). After two washes with PBS containing 1% FBS, cells were fixed and permeabilized using commercial solutions (eBioscience) for 20 min and 15 min, respectively, at room temperature.

Intracellular staining was performed by incubating permeabilized cells for 30 min with the appropriate volume of Phycoerythrin-Cy7 (PE-Cy7) anti-human IFN-γ, PE anti-human interleukin-2 (IL-2), APC anti-human TNF-α (all from eBioscience). After final washes with PBS containing 1% FBS, samples were analyzed using a flow cytometry (FCM, BD Biosciences).

For the analysis of ICS data, a standardized FlowJo (Tree Star Inc., Ashland, OR, USA) template was employed to ensure consistency across samples. This template included a gate to define the lymphocyte population based on forward scatter versus side scatter, followed by a CD3 gate to identify CD3+ lymphocytes. A CD4 and CD8 dot plot was then created, gated on CD3+ lymphocytes, with gates drawn for CD3+CD4+ and CD3+CD8+ double-positive populations. For each T-cell subset (CD3+CD4+ and CD3+CD8+), seven additional dot plots were displayed: side scatter versus IFN-γ, IL-2, TNF-α, IFN-γ & IL-2, TNF-α & IL-2, TNF-α & IFN-γ and IFN-γ & TNF-α & IL-2. Gates were set on these graphs to include positive cells (Appendix A). The results were exported from FlowJo into a spreadsheet and normalized to events per 0.1 million lymphocytes, with these normalized values referred to as the responses. The graphical representation was generated using FACSDiva software (Version 6.0A, BD Biosciences, San Jose, CA, USA) and is shown in Figure 2.

### 2.6. Quantification of HIV-1 p24, Adenovirus Vector-Binding, and Adenovirus Vector-Neutralizing Antibodies

Serum antibodies targeting HIV-1 gag and the Ad5 vector were quantified using an indirect enzyme-linked immunosorbent assay (ELISA). Briefly, 96-well microtiter plates were coated with either 200 ng/well recombinant HIV-1 HXB2 p24 or 8 × 10^8^ viral particles (VP)/well of rAd5-null and incubated overnight at 4 °C. Plates were then blocked with PBS containing 5% skimmed milk for 2 h at 37 °C, followed by five washes with PBST. Serial dilution serum samples were added to each well and incubated for 1 h at 37 °C.

Following five additional washes,100 μL of HRP-conjugated goat anti-monkey IgG antibody (1:20,000 dilution) was added to each well. Plates were incubated for another hour at 37 °C, followed by the addition of 3,3′,5,5′-tetramethylbenzidine (TMB) substrate. After 30 min of incubation at 37 °C, the reaction was stopped with 1 M H_2_SO_4_. Absorbance was measured at 450 nm, with 630 nm as the reference wavelength. The endpoint antibody titer was defined as the highest serum dilution at which the absorbance at 450 nm exceeded twice the background signal.

For Ad5-neutralizing antibody detection, serum samples were heat inactivated at 56 °C for 30 min. The inactivated sera were serially diluted (1:25 to 1:1600) and mixed with recombinant Ad5 (Ad5-luc, 4 × 10^8^ VP/mL) in 96-well plates. After 1 h of incubation at 37 °C, freshly prepared A549 cells (1 × 10^5^ cells/mL) were added to each well (100 μL/well). Twenty-four hours post-infection, luciferase activity was quantified using a GloMax luminometer (Promega, Fitchburg, WI, USA). Neutralizing antibody titers were calculated based on the reduction in luciferase activity compared to control wells.

### 2.7. Statistical Analysis

Gag-specific cellular immune responses were quantified using ELISPOT, with results reported as IFN-γ spot-forming cells (SFCs) per 10^6^ peripheral blood mononuclear cells (PBMCs). Humoral immunity against HIV p24 and Ad5 vectors was evaluated through serum antibody titers, calculated as the geometric mean titer (GMT) for binding and neutralizing antibodies. Statistical comparisons between groups were performed using descriptive statistics (mean ± SD) and unpaired Student’s *t*-tests (GraphPad Prism 9.0, GraphPad Software, Boston, MA, USA), with significance defined as *p* < 0.05.

## 3. Results

### 3.1. HIV-1 Gag Protein Expression in Response to DNA, rAd5, rSeV, and rMVA Vaccines

To validate the immunogenicity of the four vaccine platforms, we first assessed HIV-1 gag protein expression using Western blotting. The HEK293T cells infected with rAd5, rSeV, or rMVA, or transfected with the DNA vaccine, exhibited a distinct band at 55 kDa corresponding to the gag protein. No such band was observed in empty vector controls or mock-treated cells (Figure 3A). Furthermore, densitometric analysis demonstrated comparable gag protein expression levels across all four vaccine platforms, with no statistically significant differences observed (*p* > 0.05, Figure 3B). These results confirmed that all four vaccine platforms effectively expressed the HIV-1 gag protein, supporting their use in subsequent immunization studies.

### 3.2. Assessment of HIV-1 Gag-Specific Cellular Immune Responses Following Sequential Vaccination

To evaluate the efficacy of the quadruple heterologous vaccination strategy, HIV-1 gag-specific IFN-γ responses were quantified by ELISPOT assays. Building on our previous findings with a triple-vector regimen (DNA, rAd5 and rSeV) [24], we introduced a fourth vaccine rMVA when CTL responses in the triple-vector group declined below 200 SFCs/10^6^ PBMCs. A second round of vaccinations was administered to sustain long-term immunity. The four-vector regimen demonstrated substantially greater magnitude and duration of immune responses compared to the single animal in the two-vector cohort (Figure 4). While formal statistical comparison was precluded by the two-vector group’s sample size (*n* = 1), qualitative differences were evident: the four-vector group maintained responses >500 SFCs/10^6^ PBMCs for 69 weeks versus 12 weeks in the two-vector macaque. Control group data, which showed no detectable responses, were omitted for clarity.

As shown in Figure 4, the administration of rMVA at weeks 206 and 236 triggered a robust immune response in macaques previously immunized with the prime–boost–boost regimen, peaking at 3339 ± 1011 SFCs/10^6^ PBMCs (week 252). Notably, immune responses exceeding 500 SFCs/10^6^ PBMCs were maintained for 23 weeks following the initial rMVA dose and for 28 weeks following the second dose. In contrast, a macaque pre-inoculated with PBS exhibited minimal responses to rMVA, with only a modest increase after the second dose. These responses remained significantly lower than those in the four-vector group and were relatively short-lived, underscoring the importance of prior immunization in eliciting robust and sustained immunity.

At week 282, the four-vector group received a second round of vaccinations, beginning with rAd5. This intervention triggered a rapid surge in immune responses, peaking at 3194 SFCs/10^6^ PBMCs (week 284) within two weeks. Elevated responses persisted for 52 weeks (weeks 284–336), with a second peak (week 338) observed 2 weeks after the second rAd5 dose at week 336. Remarkably, the response remained above 500 SFCs/10^6^ PBMCs for an extended period of 69 weeks (weeks 336–405). In comparison, the two-vector group, which received rMVA followed by rAd5, exhibited a weaker immune response, further highlighting the superiority of the four-vector regimen in sustaining long-term immunity.

### 3.3. Breadth of HIV-1 Gag-Specific Cellular Immune Responses Following Sequential Vaccination

To assess the breadth of immune responses, we analyzed HIV-1 gag-specific IFN-γ responses against five gag peptide sub-pools (AB, CD, EF, GH and IJ) using ELISPOT assay. Following each vaccination, all macaques exhibited broad gag-specific responses, with reactivity against four or five sub-pools at peak response times (Figure 5). However, the magnitudes and specificity of responses varied among individuals. For instance, macaque 98R0053 showed a dominant response to the CD sup-pool, while macaques 98R0035 and 00R0027 exhibited stronger reactivity against the EF and AB sub-pools, respectively. In contrast, responses in macaques 00R0019, 98R0013, and 98R0037 were primarily focused on two to three sub-pools (Figure 5). These findings highlight the heterogeneity of immune responses induced by the sequential vaccination strategy.

Following each peak in gag-specific responses, the number of reactive sub-pools gradually declined, eventually stabilizing at a baseline level as the magnitude of responses diminished. However, since our analysis was conducted at the sub-pool level rather than individual peptides, we were unable to precisely determine the number of epitopes or identify immunodominant regions within the gag protein. This constraint underscores the necessity of subsequent investigations to perform finer epitope mapping to fully characterize the breadth of vaccine-induced immune responses.

### 3.4. Comprehensive ICS Profiling of Gag-Specific T Cell Response

To further characterize the quality of gag-specific T cell responses, we performed multicolor ICS assays on PBMC samples collected at two points corresponding to near peak and peak ELISPOT responses, weeks 248 and 284. Flow cytometry analysis revealed the presence of both CD4+ and CD8+ gag-specific T cells in all vaccinated macaques (Figure 6 and Appendix A). Multifunctional T cells co-expressing IFN-γ and IL-2, IFN-γ and TNF-α, IL-2 and TNF-α or all three cytokines were detected in both CD4⁺ and CD8⁺ populations. Notably, the four-vector group exhibited significantly higher frequencies of T cells secreting one cytokine or T cells secreting two and three cytokines simultaneously within both CD4+ and CD8+ populations compared to the two-vector group, except for T cells secreting TNF-α at weeks 248 and 284. These results demonstrate that the quadruple heterologous regimen not only enhances the magnitude but also improves the quality of cellular immune responses.

Due to the inherently lower sensitivity of the ICS assay compared to the ELISPOT assay, the magnitude of immune response detected by ICS was consistently lower. While the ELISPOT assay measures IFN-γ secretion from all lymphocytes, ICS specifically distinguishes cytokine production in CD4+ T and CD8+ T cell subsets. This difference in resolution explains why IFN-γ responses were detected by ELISPOT in the two-vector group but not by ICS in CD8+ T cells. Importantly, the four-vector group exhibited significantly enhanced CD8+ T cell responses and IFN-γ production compared to the two-vector group, further underscoring the superiority of the sequential heterologous vaccination strategy in eliciting robust cellular immunity.

### 3.5. Evaluation of Antibody Responses to HIV-1 p24 and Adenovirus in Vaccinated Macaques

We next evaluated humoral immune responses by measuring serum antibodies against HIV-1 p24 and Ad5 vector. Baseline p24 antibody titers were low (~10^3^) but increased following vaccination, stabilizing after an initial decline (Figure 7A). Both the four-vector and two-vector groups showed similar p24 antibody levels, with a sharp increase (~4-fold) observed after the second rAd5-HIVgag vaccination at week 336. However, this peak was transient, contrasting with the sustained cellular immune responses. These findings suggest that the multi-vector regimen primarily enhances cellular rather than humoral immunity.

Despite repeated immunizations (weeks 282–405), Ad5-binding antibody titers in the rAd5-HIVgag serum remained relatively low (Figure 7B). In the four-vector group, titers fluctuated between 10^2^ and 10^3^, while the two-vector group maintained background levels until week 282, when the first rAd5 vaccination of the second round was administered. Following the second rAd5-HIVgag immunization at week 336, Ad5-binding antibody titers increased markedly by week 338, with a 4-fold rise in the four-vector group (200 to 800 in three of four animals) and a 2-fold rise in the two-vector group (200 to 400). Titers remained stable thereafter. Throughout the study, the four-vector group consistently exhibited higher Ad5-binding antibody titers than the two-vector group, suggesting that early rAd5-HIVgag vaccination primes the immune system for stronger vector-specific antibody responses upon reimmunization.

In the two-vector group, Ad5-neutralizing antibody titers remained at background levels throughout the study (Figure 7C). In contrast, the four-vector group showed significantly higher neutralizing antibody titers, ranging between 10^2^ and 10^3^. After the second rAd5-HIVgag dose at week 336, Ad5-neutralizing titers exhibited minimal variation. The stable Ad5-neutralizing antibody (NAb) titers following the second Ad5 vaccination indicate that pre-existing vector immunity remained consistent, suggesting the primary Ad5 immunization effectively primed the humoral response without eliciting further NAb amplification upon boosting. This observation implies minimal interference by vector-specific immunity with subsequent antigen delivery. Notably, while Ad5-binding antibodies increased post-second vaccination (week 336), reflecting active immune recognition of the vector, this did not correlate with a proportional rise in NAbs that could compromise vaccine efficacy. Critically, despite stable NAb levels, the heterologous regimen maintained robust cellular immunogenicity, as evidenced by a secondary ELISPOT response peak at week 338 (two weeks post-second rAd5 dose) and sustained IFN-γ⁺ T-cell responses exceeding 500 SFCs/10^6^ PBMCs for 69 weeks (weeks 336–405, Figure 4). These findings collectively demonstrate that the sequential heterologous strategy successfully balances the induction of durable cellular immunity with controlled vector-specific humoral responses.

## 4. Discussion

The COVID-19 pandemic has underscored the potential of heterologous vaccination strategies, demonstrating that combining diverse vaccine platforms can enhance immune protection against rapidly evolving pathogens [25,26]. However, most studies have been limited to short-term observations, leaving the long-term efficacy and safety of such approaches largely unexplored [27]. In contrast, this study provides a comprehensive longitudinal evaluation of sequential prime–boost–boost strategy using four heterologous HIV vaccines in rhesus macaques over a decade.

Recent research has increasingly focused on developing vaccines capable of eliciting both cellular immune responses and antibody production [28,29]. While clinical trials of prophylactic vaccines have demonstrated that cytotoxic T-cell responses can reduce viral load, these responses alone are insufficient to prevent HIV infection. This limitation is likely due to the transient nature of vaccine-induced cellular immunity, which fails to provide durable protection. A critical challenge in HIV vaccine development is thus the induction of long-lasting immune responses. Our study addresses this challenge by demonstrating that the four-vector regimen not only elicits stronger initial immune responses but also sustains them over extended periods. Repeated immunizations with the rAd5-HIVgag vaccine reactivated high-level gag-specific cellular immunity, as evidenced by sustained ELISPOT responses lasting 52 to 69 weeks. These findings align with recent advances in COVID-19 vaccine research, where heterologous boosting strategies have been shown to induce more durable humoral and cellular immune responses compared to homologous regimens [27,30]. Similarly, our study highlights the importance of repeated heterologous boosting in sustaining long-term immunity, a strategy that could be particularly valuable for HIV, where lifelong immune protection is required.

Findings from the Step Merck Adenovirus 5 (MRKAd5) HIV-1 gag/pol/nef study demonstrate that gag-specific responses are associated with reduced viremia, highlighting their role in controlling viral load during chronic infection [31,32]. Further evidence suggests that vaccine-induced T cells can significantly lower plasma viremia, particularly in individuals targeting multiple gag peptides, who exhibit reduced viral loads [33,34]. The induction of potent cellular immune responses, particularly CD8+ T cells, is a critical component of an effective HIV vaccine. In our study, the four-vector regimen induced higher frequencies of multifunctional CD4+ and CD8+ T cells secreting multiple cytokines simultaneously compared to the two-vector animal group, underscoring the superiority of the sequential heterologous vaccination strategy in eliciting robust cellular immunity. These results support the growing consensus that an effective HIV vaccine must engage both cellular and humoral arms of the immune system, with CD8+ T cells playing a pivotal role in viral control [3].

Our findings are consistent with the growing body of evidence supporting the efficacy of heterologous prime–boost strategies, where initial DNA vaccine immunization is followed by a booster using a viral vector or recombinant–protein vaccine to a induce stronger HIV-specific immune response [35]. This approach has been validated in non-human primate models, demonstrating that primary immunization with SIV DNA vaccines, followed by adenovirus vector boosters, effectively reduces viral loads and slows disease progression [36].

Vector-based vaccines are highly effective at inducing cellular immune responses. However, the immunogenicity of the vector itself often limits their repeated application [37,38]. To address this challenge, sequential administration of multiple vectors can be employed. When immune responses induced by one vaccine decline, subsequent vaccines can reinforce and prolong the immunotherapeutic effect. In a previous study, we demonstrated that prime–boost–boost strategies using three vectors expressing the HIV-1 gag gene could elicit potent and specific immune responses [24]. The four vaccines used in this study utilize distinct vectors to deliver either identical or varied exogenous HIV genes. These vectors introduce foreign genes into somatic cells in vivo, where the expression of the encoded antigens triggers heterologous protein-specific immune responses, particularly CTL responses [35]. While any vector proven safe and effective in delivering foreign genes could serve as a vaccine platform, immune responses and antibodies specific to the vector itself—induced by vector-encoded antigens—can hinder subsequent vaccinations. Sequential and repeated administration of four-vector HIV vaccines overcomes this limitation by using different vectors to reinforce immune responses initiated by prior vaccinations, thereby enhancing and extending the duration of immunity.

The success of heterologous prime–boost strategies in combating other infectious diseases, such as COVID-19, provides a strong rationale for their application in HIV vaccine development. Recent studies have shown that heterologous vaccination regimens can enhance both humoral and cellular immune responses, offering a promising alternative to conventional single-platform vaccines [30,39]. Notably, a combination of adenovirus vector vaccines with mRNA or inactivated vaccines has significantly enhanced immune protection against variants like Omicron [40]. However, most COVID-19 studies have been limited by relatively brief observation periods (typically ranging from days to months), constraining our ability to fully assess the long-term safety and efficacy profiles of these vaccination strategies. In contrast, our study provides a comprehensive longitudinal evaluation of a sequential heterologous prime–boost–boost strategy over a decade.

Despite the valuable insights provided by our study, several limitations must be acknowledged. First, the use of rhesus macaques, which are not susceptible to HIV infection, restricts our ability to assess the protective efficacy of the vaccine regimen. Future studies should evaluate this strategy in SIV-infected rhesus macaques, as performed in a separate study (unpublished). Second, the small sample size of the two-vector vaccine group (*n* = 1) and control groups (*n* = 2 for PBS controls) may limit the statistical power to detect subtle differences between experimental and control cohorts, potentially affecting the generalizability of our findings. Future investigations should prioritize larger cohorts to validate the robustness of the observed immune dynamics. The limited sample size in the two-vector group (*n* = 1) precludes definitive comparative conclusions, although observed response patterns align with established principles of heterologous prime–boost immunology. Third, the absence of baseline serum antibody data between weeks 87 and 206 may have contributed to the observed similarities in p24-specific antibody responses between the four-vector and two-vector groups. Future research should focus on elucidating the dynamics of humoral immune responses in heterologous vaccination regimens.

## 5. Conclusions

In conclusion, our study demonstrates that sequential and repeated immunization with four heterologous vector-based HIV vaccines can induce potent and prolonged gag-specific cellular immune responses in rhesus macaques. This strategy holds significant promise for the development of therapeutic and prophylactic HIV vaccines, offering a novel approach to enhance and sustain immune protection against this persistent virus. As the global fight against HIV/AIDS continues, exploring innovative vaccination strategies, such as heterologous prime–boost regimens, will be crucial for advancing vaccine development and immunotherapy.

## Figures and Tables

**Figure 1 vaccines-13-00338-f001:**
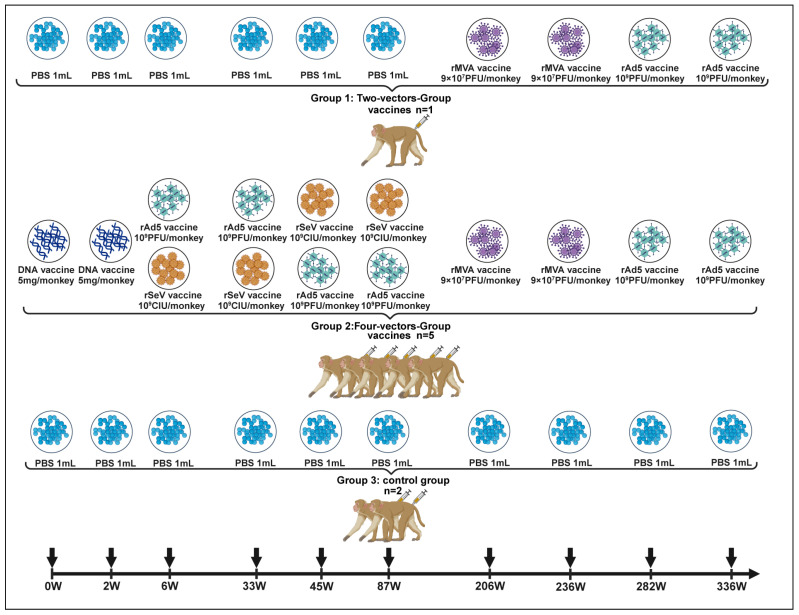
Immunization schedule and experimental design for rhesus macaques. Schematic representation of the immunization schedule for rhesus macaques. Animals were divided into three groups: four-vector vaccine group (*n* = 5), two-vector vaccine (*n* = 1), and PBS control group (*n* = 2). The four-vector group received sequential vaccinations with DNA (weeks 0 and 2), rAd5/rSeV (weeks 6, 33, 45, and 87), rMVA (weeks 206 and 236), and a second round of rAd5 (weeks 282 and 236). The two-vector group received PBS during the initial phase, followed by rMVA (weeks 206 and 236) and rAd5 (weeks 282 and 236). The control group received PBS at all time points. Arrows indicates vaccination time points. Abbreviations: PBS, sterile phosphate-buffered saline; DNA, DNA vaccine; rAd5, recombined Adenovirus type 5 vaccine; rSeV, recombined Sendai virus vaccine; rMVA, Modified Vaccinia Virus Ankara vaccine.

**Figure 2 vaccines-13-00338-f002:**
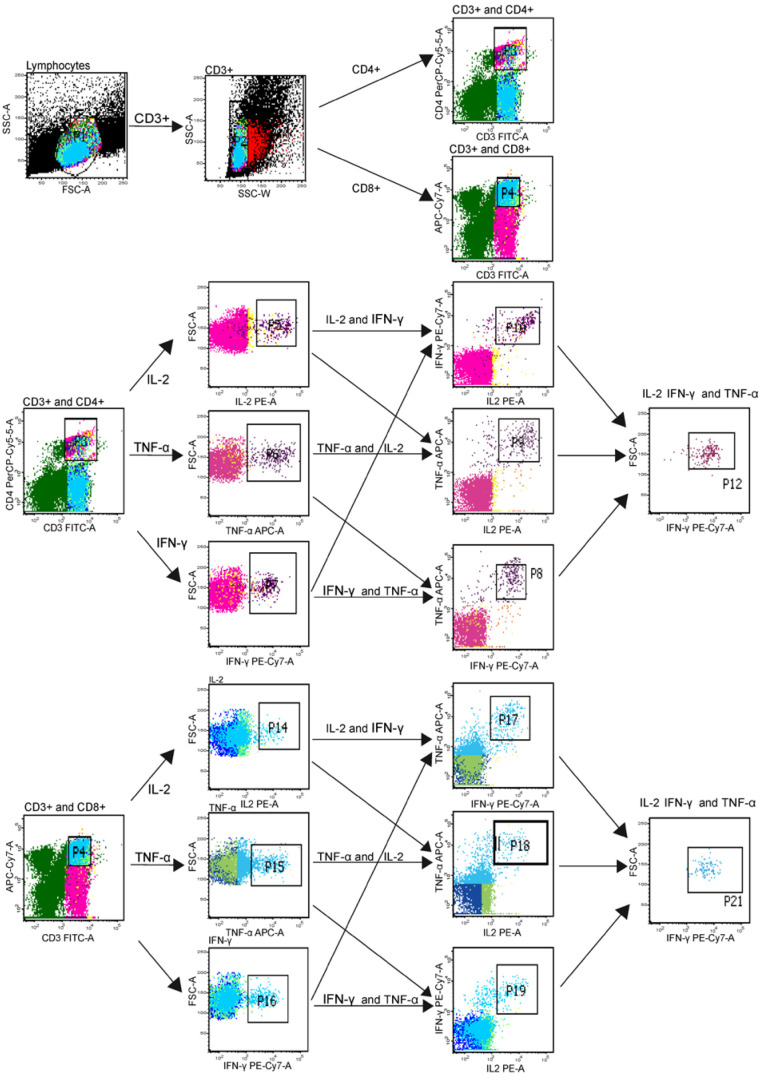
Gating strategy for multifunctional T-cell response analysis using six-color intracellular cytokine staining (ICS). Representative data from a four-vector vaccinated macaque illustrate the sequential gating protocol to quantify HIV gag-specific CD8⁺ and CD4⁺ T-cell responses. Lymphocytes were first gated on FSC-A/SSC-A plots, followed by CD3⁺ T-cell identification and subset stratification into CD4⁺ or CD8⁺ populations. Cytokine-positive gates (single, dual or triple-positive) were defined using mock-stimulated controls as a reference, with synchronized gating criteria applied uniformly across all samples.

**Figure 3 vaccines-13-00338-f003:**
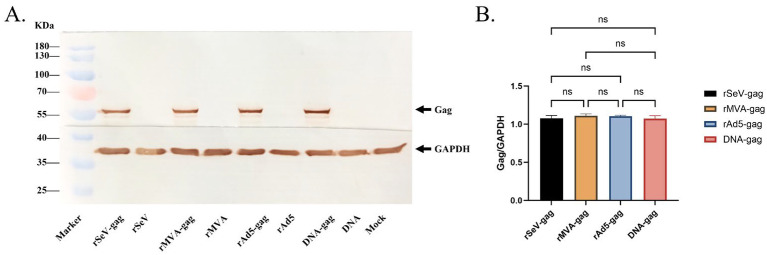
In vitro expression of HIV gag protein by four vaccine platforms. (**A**) Western blot analysis of HIV-1 gag protein expression in HEK293T cells infected with rAd5, rSeV, rMVA vaccines or transfected with the DNA vaccine. Cell lysates were probed with anti-HIV p24 and anti-GAPDH antibodies. GAPDH served as a loading control. Mock cells and empty vector controls were included to confirm specificity. The positions of the gag protein (55 kDa) and GAPDH are indicated by arrows. (**B**) Densitometry analysis of gag protein expression levels. Gray scale intensity ratios (gag/GAPDH) were calculated to quantify relative expression. No significant differences in gag expression levels were observed among the four vaccine platforms. ns, not significant.

**Figure 4 vaccines-13-00338-f004:**
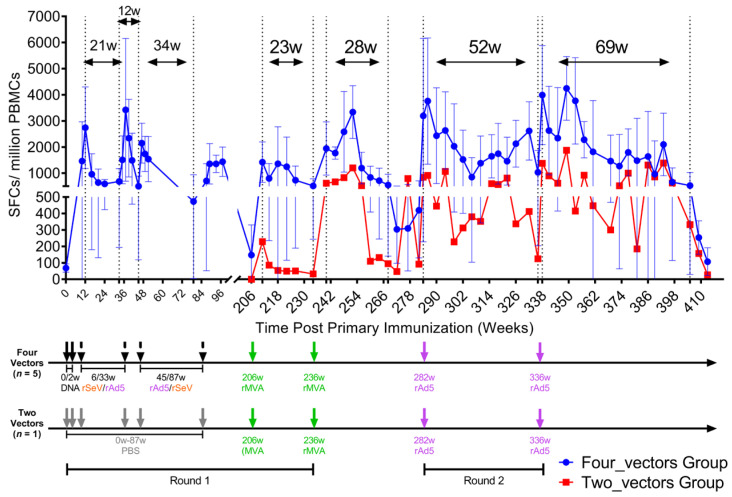
Magnitude of HIV-1 gag-specific cellular immune responses following sequential vaccination. HIV-1 gag-specific IFN-γ responses to peptide pools were measured by ELISPOT assay in rhesus macaques immunized with two-vector or four-vector regimens. Results are expressed as spot-forming cells (SFCs) per 10^6^ PBMCs. Arrows indicate vaccination time points. Comparative ELISPOT trajectories between the four-vector cohort (*n* = 5) and the single two-vector animal. Sustained responses >500 SFCs/10^6^ PBMCs persisted 69 weeks in four-vector animals versus 12 weeks in the two-vector subject. Data from the PBS control group, which showed no detectable responses, are omitted for clarity. Vaccines: black represents the DNA vaccine, orange the rSeV vaccine, green the rMVA vaccine, purple the rAd5 vaccine, and gray the PBS.

**Figure 5 vaccines-13-00338-f005:**
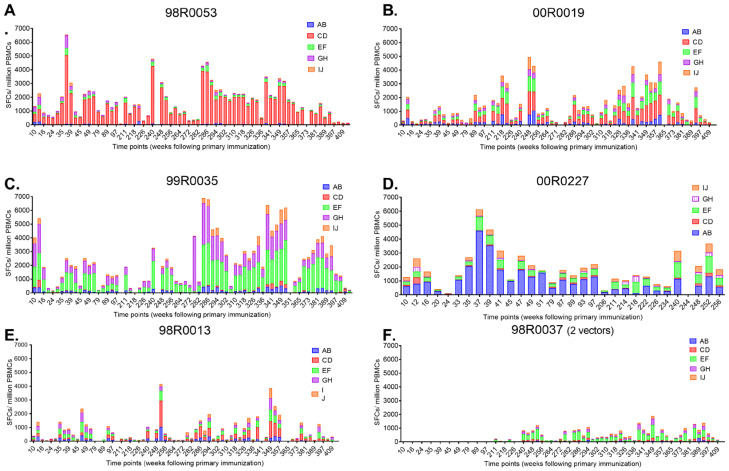
Breadth of HIV-1 gag-specific cellular immune responses against peptide sub-pools. HIV-1 gag-specific IFN-γ responses against peptide sub-pools (AB, CD, EF, GH, IJ) in four-vector (*n* = 5) and two-vector (*n* = 1) groups. (**A**–**F**) represent individual rhesus macaques, panel titles indicate animal IDs, and grouping details are provided in Methods Section 2.3. Immunization schedule is identical to Figure 4. Responses are shown as spot-forming cells (SFCs) per 10^6^ PBMCs. Individual macaques exhibited varying patterns of reactivity, with some showing dominant responses to specific sub-pools (e.g., 98R0053 to CD, 98R0035 to EF, and 00R0027 to AB). Responses in the two-vector group (98R0037) were focused on fewer sub-pools.

**Figure 6 vaccines-13-00338-f006:**
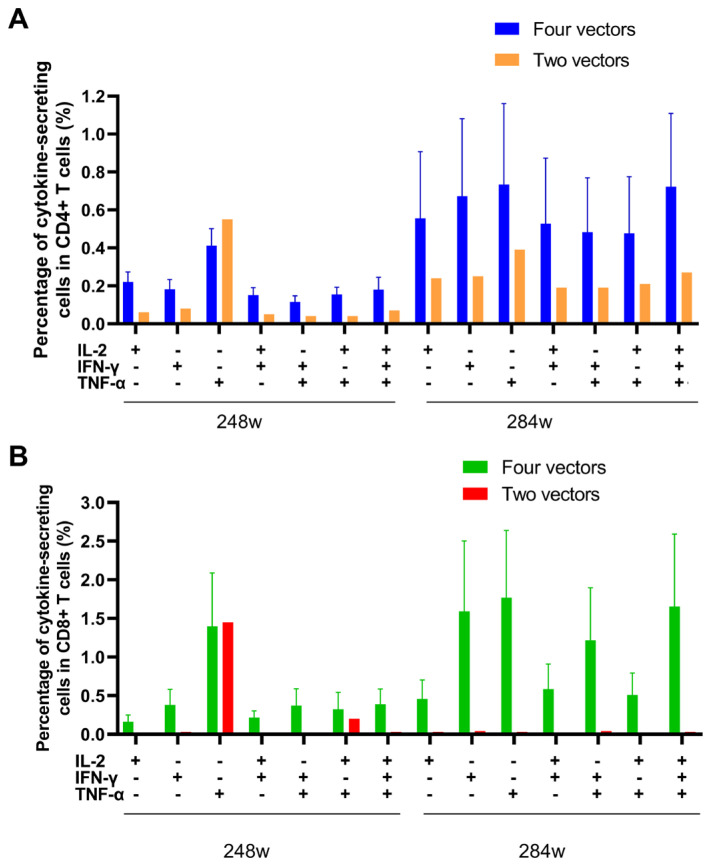
Frequency of the total HIV gag T-cell response per subset by ICS. A graphical representation of the frequency of the total CD4 (**A**) or CD8 (**B**) T-cell response to HIV gag for each of the subsets is displayed. The three functional markers (IFN-γ, IL-2 and TNF-α) measured are listed along the x axis, along with each of the 7 subsets. All responses shown are mock subtracted (i.e., HIV gag result minus DMSO (negative stimulation) result).

**Figure 7 vaccines-13-00338-f007:**
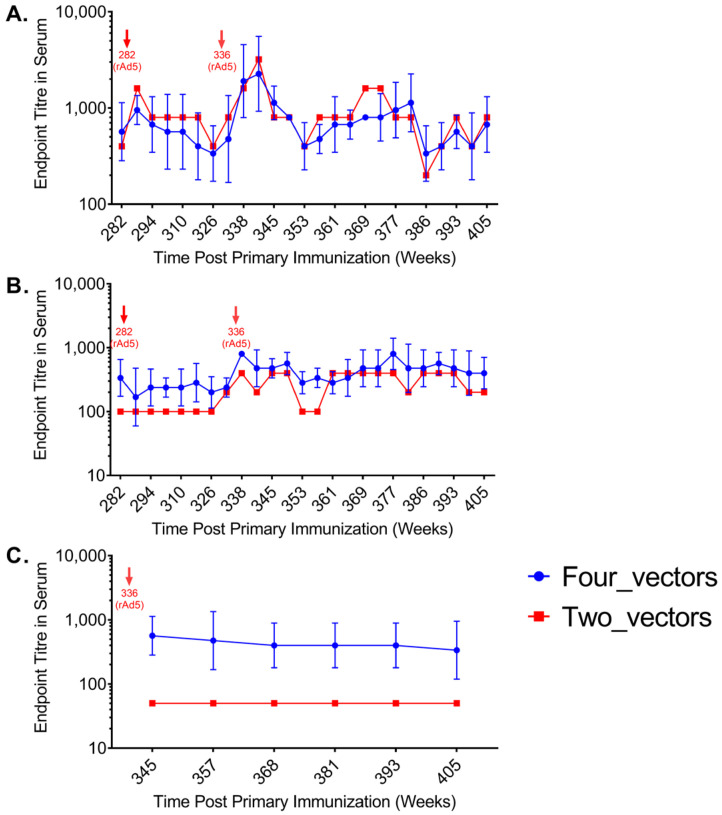
Temporal profile of antibody responses induced by sequential vaccination. Antibody responses to HIV-1 p24 and Ad5 vector in rhesus macaques immunized with two-vector or four-vector regimens. (**A**) HIV-1 p24-specific IgG titers. (**B**) Ad5-binding antibody titers. (**C**) Ad5-neutralizing antibody titers. Titers are expressed as the mean for each group. The four-vector group showed higher Ad5-binding and neutralizing antibody titers compared to the two-vector group, while p24-specific IgG titers were similar between the two groups. Arrows indicate vaccination time points.

## Data Availability

All data related to this study are included in this article.

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
