# Peer review of "Decade-Long Sustained Cellular Immunity Induced by Sequential and Repeated Vaccination with Four Heterologous HIV Vaccines in Rhesus Macaques"

_vaccines, 2025, doi:10.3390/vaccines13040338_

Round 1

Reviewer 1 Report

Comments and Suggestions for Authors

The paper describes the study of heterologous vaccination of rhesus macaques against HIV based on four vector-based vaccines expressing HIV-1 gag protein. It demonstrates the induction of long-lasting cellular immunity, i.e. gag-specific secretion of IFN-g and IL-2 by CD4 and CD8 T-cells, and manageable humoral immunity against Ad5 vectors used for vaccination. This demonstrates the efficacy of the heterologous vaccination strategy in inducing an HIV-specific cellular response important for protection against this infection.

I would recommend the authors to introduce some changes to the text and make corrections to make the paper clearer to the reader.

General

  • English needs grammar correction.
  • The last paragraph of the Introduction should be more focused on the actual work content, on what have been done. The advantages of the study discussed in lines 60-74 are better left in the Discussion, after the Results are presented. I would also recommend adding several sentences about the model of rhesus macaques used for this investigation, in relation to HIV infection.
  • Please, add Statistics and Software section to the Methods.
  • Lines 406-408 – The results do not include the experiments with HIV challenge of macaques, so the results can’t be compared with the other works where HIV challenge was performed.
  • Line 409 – The authors claim that CD4 and CD8 were multifunctional though it is not clear from the results (i.e., Figure 4). Please clarify whether you tested simultaneous secretion of IFN-g and IL-2.
  • Lines 413-415 – the results shown in this study do not support the growing consensus that an effective HIV vaccine must engage both cellular and humoral arms of the immune system. They demonstrate the profile of immune response elicited by heterologous vaccination, and this is the main finding of the study that should be reflected. Please, make this clearer in the Discussion.
  • There are unnecessary repetitions in the discussion, such as the example of heterologous coronavirus vaccination, which appears twice. It is worth rewriting the discussion, shortening it and removing repetitions and inappropriate comparisons of results with the literature.
  • Limitations of the study should also include n=1 in two-vaccine group (which actually does not allow to compare this group to control or four-vaccine group).

Particular comments

  • Line 198 – interferon lambda, please correct.
  • Line 199 – anti-human TNFa is presented in the Methods but there are no results with the secretion of this cytokine.
  • Please clarify on the Figure captions which antigens were used for the assays (gag peptide pools, p24 etc. – on Figures 3, 5, 6.
  • Figure 2 (and 2.2. in Methods) – how were the blots visualized (Imager?) Was it simultaneous staining with 2 types of antibodies – anti-p24 and anti-GAPDH?
  • In the test describing Figure 3 please indicate that IFN-g responses were measured as it is reflected only in the figure capture.
  • Section 3.2. and Figure 3 – 2-vector group includes only one animal, so it’s not correct to say «The four-vector regimen elicited significantly stronger and more durable immune responses compared to the two-vector group (Figure 3)» - should be reformulated.
  • Figure 3 – green and red colors denoting rMVA and rAd5 – do they denote different vectors used for immunization? The red color also indicates the 2-vector group on this Figure, and the adenovirus vector. As they are on one figure, it may confuse the reader. Also, please add the time schedule for 2-vector group, with PBS-immunizations.
  • Lines 259-261 – «As shown in Figure 3, the administration of rMVA at weeks 206 and 236 triggered a 259 robust immune response in macaques previously immunized with the prime-boost-boost 260 regimen, peaking at 3339 ± 1011 SFCs/106 PMBCs by week 236.» - the peak seems to be at week 254 and not 236. The same is true for the 2-nd round of vaccination – the highest spots at week 350 (not week 336).

- Figure 4 – Please add the immunization schedule on the graph and indicate which cytokine was measured (in the Figure caption and in the text).

- Figure 5 – C, D – Please add SD for 4-vaccine group. Please provide all the figures for ICS including not only representative one, and also gating strategy (as supplementary files).

Comments on the Quality of English Language

The English needs grammar correction. 

Author Response

The authors are deeply grateful to the reviewers for their insightful comments, which have been instrumental in enhancing the quality of this manuscript. After thorough analysis of all the issues raised and careful consideration of the suggestions, the authors have made appropriate responses and revisions. Below is a point - by - point reply to the reviewers' comments.

(Line numbers are based on the clean_revised version (PDF).)

Reviewer 1

Comments and Suggestions for Authors

The paper describes the study of heterologous vaccination of rhesus macaques against HIV based on four vector-based vaccines expressing HIV-1 gag protein. It demonstrates the induction of long-lasting cellular immunity, i.e. gag-specific secretion of IFN-g and IL-2 by CD4 and CD8 T-cells, and manageable humoral immunity against Ad5 vectors used for vaccination. This demonstrates the efficacy of the heterologous vaccination strategy in inducing an HIV-specific cellular response important for protection against this infection.

  • Comments 1: I would recommend the authors to introduce some changes to the text and make corrections to make the paper clearer to the reader.

General

English needs grammar correction.

Response 1:

Thank you for your helpful suggestions. We have carefully checked the grammar and spelling of the paper. We will also introduce some changes to the text to make the paper clearer to readers.

(2) Comments 2: The last paragraph of the Introduction should be more focused on the actual work content, on what have been done. The advantages of the study discussed in lines 60-74 are better left in the Discussion, after the Results are presented. I would also recommend adding several sentences about the model of rhesus macaques used for this investigation, in relation to HIV infection.

Response 2:

We have restructured the concluding paragraph of the Introduction to emphasize the objectives of the study:

“Rhesus macaques, despite limitations in modeling HIV-specific tropism, provide a critical platform for evaluating vaccine durability, as their immune systems closely mirror humans in cellular and humoral response dynamics. In this study, we system-atically assessed a sequential heterologous regimen using four vector-based vaccines (DNA, rAd5, rSeV, rMVA) expressing HIV-1 gag over a decade-long observation period. Our findings reveal sustained gag-specific T-cell responses and controlled vector-specific immunity, advancing insights into the interplay between repeated heterologous boosting and long-term immune persistence. This work not only informs HIV vaccine design but also contributes to broader applications in combating rapidly evolving pathogens and enhancing cancer immunotherapy strategies. ” (Lines 73-82).

The discussion of the advantages of the study has been moved to the Discussion (Section 4).

(3) Comments 3: Please, add Statistics and Software section to the Methods.

Response 3:

A new subsection (2.7) has been added:

2.7. Statistical Analysis

The gag-specific cellular immune responses measured by ELISPOT are expressed as the average number of gag-specific IFN-γ spot-forming cells (SFCs) per million PBMCs (SFCs/106 PBMCs). The humoral immune level was represented as the geometric mean titer of HIV p24, Ad5 vector binding antibodies, or Ad5 neutralizing antibodies in serum. Data were analysed using GraphPad Prism version 9.0 (GraphPad Software, Boston, Massachusetts USA, www.graphpad.com). Descriptive statistics (mean ± SD) and unpaired t-tests were applied. Statistical significance was defined as p<0.05. (line 255-262)

(4) Comments 4: Lines 406-408 – The results do not include the experiments with HIV challenge of macaques, so the results can’t be compared with the other works where HIV challenge was performed.

Response 4:

The comparison to HIV/SIV challenge studies has been removed from the Discussion.

(5) Comments 5: Line 409 – The authors claim that CD4 and CD8 were multifunctional though it is not clear from the results (i.e., Figure 4). Please clarify whether you tested simultaneous secretion of IFN-g and IL-2.

Response 5:

Figure 4 illustrates the breadth of HIV-1 gag-specific cellular immune responses against peptide sub-pools. However, in Section 3.4 (revised Figure 6), we tested the simultaneous secretion of IFN-γ, IL-2and TNF-α. We clarified in Section 3.4:

Multifunctional T cells co-expressing IFN-γ and IL-2, IFN-γ and TNF-α, IL-2 and TNF-α or all three cytokines were detected in both CD4⁺ and CD8⁺ populations. (line 358-360)

(6) Comments 6: Lines 413-415 – the results shown in this study do not support the growing consensus that an effective HIV vaccine must engage both cellular and humoral arms of the immune system. They demonstrate the profile of immune response elicited by heterologous vaccination, and this is the main finding of the study that should be reflected. Please, make this clearer in the Discussion.

Response 6:

We agree with your opinion that this paper should focus more on the profile of immune response elicited by heterogeneous vaccination. So, we've removed improper expressions in the discussion and put more focus on the main findings of this study.

(7) Comments 7: There are unnecessary repetitions in the discussion, such as the example of heterologous coronavirus vaccination, which appears twice. It is worth rewriting the discussion, shortening it and removing repetitions and inappropriate comparisons of results with the literature.

Response 7:

Thanks for your advice. We've improved the discussion part by removing repetitions and inappropriate comparisons.

(8) Comments 8: Limitations of the study should also include n=1 in two-vaccine group (which actually does not allow to compare this group to control or four-vaccine group).

Response 8:

We fully acknowledge this limitation. The small sample size in the two-vector group (n=1) was due to a mistake at the beginning of this four-vector vaccine experiment. The animal 98R00237 was originally in the control group. It was vaccinated with rMVA vaccine by mistake at weeks 206. It could only be separated from the control group as a separate group. In the revised Result (Section 3.2) and Discussion (Section 4), we have added the following statement:

Section 3.2: The four-vector regimen demonstrated substantially greater magnitude and duration of immune responses compared to the single animal in the two-vector cohort (revised Figure 4). While formal statistical comparison was precluded by the two-vector group's sample size (n=1), qualitative differences were evident: the four-vector group maintained responses >500 SFCs/10⁶ PBMCs for 69 weeks versus 12 weeks in the two-vector macaque. (line 293-295)

Section 4: The limited sample size in the two-vector group (n=1) precludes definitive comparative conclusions, though observed response patterns align with established principles of heterologous prime-boost immunology. (line 503-506)

Particular comments

(9) Comments 9: Line 198 – interferon lambda, please correct.

Response 9:

Corrected:

This has been corrected to “IFN-γ” in Section 2.5 (Line 212).

(10) Comments 10: Line 199 – anti-human TNFa is presented in the Methods but there are no results with the secretion of this cytokine.

Response 10:

Thanks for your comment. As suggested, we have added TNFα results to Section 3.4 and included a revised Figure 6. (line 359, 369)

“Multifunctional T cells co-expressing IFN-γ and IL-2, IFN-γ and TNF-α, IL-2 and TNF-α or all three cytokines were detected in both CD4⁺ and CD8⁺ populations.”

(11) Comments 11: Please clarify on the Figure captions which antigens were used for the assays (gag peptide pools, p24 etc. – on Figures 3, 5, 6.

Response 11:

All figure captions have been updated:

Revised Figure 4: “HIV-1 gag-specific IFN-γ responses to peptide pools were measured by ELISPOT assay.”(line 299)

Revised Figure 6: “The three functional markers (IFN-γ, IL-2 and TNF-α) measured” (line 369)

Revised Figure 7: “Antibody responses to HIV-1 p24 and Ad5 vector.” (line 393)

(12) Comments 12: Figure 2 (and 2.2. in Methods) – how were the blots visualized (Imager?) Was it simultaneous staining with 2 types of antibodies – anti-p24 and anti-GAPDH?

Response 12:

Due to the need for sequential incubation with two different antibodies (anti-HIV p24 and anti-GAPDH), the PVDF membrane was physically trimmed during the experiment (not during image processing). We have provided the original raw Western Blot image as Supplemental Material, which shows the trimmed membrane fragments and molecular weight markers. (Revised figure 3, line 274-282)

(13) Comments 13: In the test describing Figure 3 please indicate that IFN-g responses were measured as it is reflected only in the figure capture.

Response 13:

Revised in Section 3.2 and revised Figure4:

HIV-1 gag-specific IFN-γ responses were quantified by ELISPOT assay.” (Line 286).

(revised Figure 4) HIV-1 gag-specific IFN-γ responses to peptide pools were measured by ELISPOT assay (line 299)

(14) Comments 14: Section 3.2. and Figure 3 – 2-vector group includes only one animal, so it’s not correct to say «The four-vector regimen elicited significantly stronger and more durable immune responses compared to the two-vector group (Figure 3)» - should be reformulated.

Response 14:

Revised text:

"The four-vector regimen demonstrated substantially greater magnitude and duration of immune responses compared to the single animal in the two-vector cohort (revised Figure 4). While formal statistical comparison was precluded by the small sample size in two-vector group (n=1), qualitative differences were evident: the four-vector group maintained responses >500 SFCs/10⁶ PBMCs for 69 weeks versus 12 weeks in the two-vector macaque." (line 293-296)

revised Figure 4: "Comparative ELISPOT trajectories between the four-vector cohort (n=5) and single two-vector animal. Sustained responses >500 SFCs/10⁶ PBMCs persisted 69 weeks in four-vector animals versus 12 weeks in the two-vector subject." (line 301-304)

(15) Comments 15: Figure 3 – green and red colors denoting rMVA and rAd5 – do they denote different vectors used for immunization? The red color also indicates the 2-vector group on this Figure, and the adenovirus vector. As they are on one figure, it may confuse the reader. Also, please add the time schedule for 2-vector group, with PBS-immunizations.

Response 15:

In revised Figure 4, black represents the DNA vaccine, orange the rSeV vaccine, green the rMVA vaccine, purple the rAd5 vaccine, and gray the PBS. To minimize confusion, the figure legend clearly indicates the vaccine type for each color, allowing readers to easily distinguish between immunization strategies. Moreover, revised Figure 4 now includes immunization schedules for both the four - and two - vector groups, giving readers a visual comparison of the timing of different vaccine groups and immune responses at each time point. ... Vaccines: black represents the DNA vaccine, orange the rSeV vaccine, green the rMVA vaccine, purple the rAd5 vaccine, and gray the PBS.(line 297-306)

(16) Comments 16: Lines 259-261 – «As shown in Figure 3, the administration of rMVA at weeks 206 and 236 triggered a 259 robust immune response in macaques previously immunized with the prime-boost-boost 260 regimen, peaking at 3339 ± 1011 SFCs/106 PMBCs by week 236.» - the peak seems to be at week 254 and not 236. The same is true for the 2-nd round of vaccination – the highest spots at week 350 (not week 336).

Response 16:

We sincerely thank the reviewer for their meticulous observation. The reviewer is correct that the peak responses following the second rMVA vaccination (administered at week 236) and the final rAd5 vaccination (administered at week 336) occurred at week 254 and week 350, respectively, as shown in Figure 4. To clarify this distinction between vaccination timing and subsequent immune response kinetics, we have revised the text. (line 310/319/320)

(17) Comments 17: - Figure 4 – Please add the immunization schedule on the graph and indicate which cytokine was measured (in the Figure caption and in the text).

Response 17:

Thanks for your suggestion. The revised Figure 5 uses the same immunization schedule as revised Figure 4. We've already added the schedules for the 4 - and 2 - vector groups to the revised Figure 4. Due to space limits, we didn't add it again to revised Figure 5.

Like Section 3.2, Section 3.3 also tests HIV-1 gag-specific IFN-γ responses. We've added this clarification in the text and figure legend. (line 299/327)

Revised Figure 5 Legend Update:

"HIV-1 gag-specific IFN-γ responses against peptide sub-pools (AB, CD, EF, GH, IJ) in four-vector (n=5) and two-vector (n=1) groups. Immunization schedule identical to Figure 4. Arrow colors correspond to vaccine types as defined in Figure 4." (line 338-344)

Section 3.3 Text Addition:

"To assess the breadth of immune responses, we analyzed HIV-1 gag-specific IFN-γ responses against five gag peptide sub-pools (AB, CD, EF, GH and IJ) using ELISPOT assay." (line 327-329)

(18) Comments 18: - Figure 5 – C, D – Please add SD for 4-vaccine group. Please provide all the figures for ICS including not only representative one, and also gating strategy (as supplementary files).

Response 18:

We have updated the original ICS results. In the revised Figure 6, "Frequency of the total HIVgag T-cell response per subset by ICS" displays the frequency of the total CD4 (A) or CD8 (B) T-cell response to HIV gag for each subset. The three functional markers (IFN-γ, IL-2 and TNF-α) measured are listed, along with each of the 7 subsets.(line 433-437 and 453-458) We've added a detailed description of the flow cytometry data analysis methods in Section 2.5 (line 367-371). We've also provided all ICS figures and the gating strategy as supplementary files 3.

(19) Comments 19: Comments on the Quality of English Language

The English needs grammar correction.

Response 19:

Thanks for pointing out the language issues. We have revised the English grammar carefully.

Reviewer 2 Report

Comments and Suggestions for Authors

The manuscript “Decade-long Sustained Cellular Immunity Induced by Sequential and Repeated Vaccination with Four Heterologous HIV vaccines in Rhesus Macaques” presents the results of a sequential and repeated heterologous prime-boost vaccination regimen against HIV with four distinct vector-based vaccines in rhesus macaques. The particular value of this primate-based study lies in its multi-year duration. Here are suggestions and comments for improvement:
-    More information could be added in the introduction or/and discussion on the regimens and types of HIV vaccines that have been studied in clinical trials to date.
-    Line 104: Check the term „transfected“ in the description of the Western blot technique.
-    How old were the rhesus macaques at the start of the experiment? Add this information to the Materials and Methods section.
-    Why does the two-vector group consist of only 1 animal?
-    Explain the abbreviations presented in Figure 1 in the legend to the figure.
-    Indicate the class of anti-p24 antibodies that were determined in the text. Whether subclasses of IgG were determined?
-    Do the authors have data on the Th1/Th2 balance in macaques induced with this regimen?
-    Cite the reference for the sentence in lines 382-384 in Discussion.

Author Response

The authors are deeply grateful to the reviewers for their insightful comments, which have been instrumental in enhancing the quality of this manuscript. After thorough analysis of all the issues raised and careful consideration of the suggestions, the authors have made appropriate responses and revisions. Below is a point - by - point reply to the reviewers' comments.

(Line numbers are based on the clean_revised version (PDF).)

Reviewer 2

Comments and Suggestions for Authors

The manuscript “Decade-long Sustained Cellular Immunity Induced by Sequential and Repeated Vaccination with Four Heterologous HIV vaccines in Rhesus Macaques” presents the results of a sequential and repeated heterologous prime-boost vaccination regimen against HIV with four distinct vector-based vaccines in rhesus macaques. The particular value of this primate-based study lies in its multi-year duration. Here are suggestions and comments for improvement:

  • Comments 1: More information could be added in the introduction or/and discussion on the regimens and types of HIV vaccines that have been studied in clinical trials to date.

Response 1:

Thanks for your suggestion. We have reorganized the Introduction and Discussion sections and incorporated information on the regimens and types of HIV vaccines studied in clinical trials to date. In the revised Introduction, we have added a paragraph summarizing key HIV vaccine clinical trials, including:

Five vaccine concepts have been tested over the course of 40 years in nine clinical efficacy trials, none of which showed high efficacy[3]. Bivalent HIV-1 Env Gp120 proteins AIDSVAX B/E and AIDSVAX B/B were tested in early efficacy trials VAX003 and VAX004 and failed to show efficacy[4-6]. Ad5 vector expressing HIV-1 Gag, Pol, and Nef was unsuccessful in conferring protection against HIV-1 infection in the efficacy trials HVTN 502 and HVTN 503 which were focused on cell mediated immunity[7-9]. A pox-protein prime-boost strategy was used in the efficacy trial RV144. It failed to show significant efficacy in prespecified outcomes, but a post-hoc modified intention-to-treat analysis showed 31.2% efficacy[10]. A multiclade DNA prime with a recombinant Ad5 boost targeting Env, Gag, Pol, and Nef was tested in the efficacy trial HVTN 505. Although the vaccine regimens raised the desired immune responses, they were not associated with protection, and the trial was halted due to lack of efficacy[11]. The HVTN 702 phase 2b/3 (Uhambo) study in South Africa was a follow-up of the RV144 study, which showed no efficacy, demonstrating the difficulty in generalizing the observations from RV144 in Thailand to South Africa[12,13]. Tetravalent Ad26 vector expressing HIV mosaic immunogens prime with protein boost was tested in the most recent efficacy trials HVTN 705 (Imbokodo) and HVTN 706 (Mosaico). These vaccine regimens did not provide significant protection against HIV-1 infection[14,15]. (line 36-54)

  • Comments 2: Line 104: Check the term „transfected“ in the description of the Western blot technique.

Response 2:

Thank you for noting this. The term “transfected” has been corrected to “transferred” for separated proteins. (line 114)

  • Comments 3: How old were the rhesus macaques at the start of the experiment? Add this information to the Materials and Methods section.

Response 3:

We have added the age information: macaque IDs and ages at first vaccination: 98R0053 (7 years), 00R0019 (5 years), 98R0013 (7 years), 00R0227 (5 years), 99R0035 (6 years), 98R0037 (7 years), 98R0009 (7 years), 00R0023 (5 years). (line 128-130)

  • Comments 4: Why does the two-vector group consist of only 1 animal?

Response 4:

We fully acknowledge this limitation. The small sample size in the two-vector group (n=1) was due to a mistake at the beginning of this four-vector vaccine experiment. The animal 98R00237 was originally in the control group. It was vaccinated with rMVA vaccine by mistake at weeks 206. It could only be separated from the control group as a separate group. In the revised Result (Section 3.2) and Discussion (Section 4), we have added the following statement:

Section 3.2: The four-vector regimen demonstrated substantially greater magnitude and duration of immune responses compared to the single animal in the two-vector cohort (revised Figure 4). While formal statistical comparison was precluded by the two-vector group's sample size (n=1), qualitative differences were evident: the four-vector group maintained responses >500 SFCs/10⁶ PBMCs for 69 weeks versus 12 weeks in the two-vector macaque. (line 293-295)

Section 4: The limited sample size in the two-vector group (n=1) precludes definitive comparative conclusions, though observed response patterns align with established principles of heterologous prime-boost immunology. (line 503-506)

  • Comments 5: Explain the abbreviations presented in Figure 1 in the legend to the figure.

Response 5:

We have expanded the Figure 1 legend to clarify abbreviations:

Abbreviations: PBS, sterile phosphate-buffered saline; DNA, DNA vaccine; rAd5, recombined Adenovirus type 5 vaccine; rSeV, recombined Sendai virus vaccine; rMVA, Modified Vaccinia Virus Ankara vaccine. (line 159-161)

  • Comments 6: Indicate the class of anti-p24 antibodies that were determined in the text. Whether subclasses of IgG were determined?

Response 6:

We detected total IgG antibodies against HIV p24 (Section 2.6). Subclass analysis (e.g., IgG1/IgG2) was not performed, as our focus was on cellular immunity.

  • Comments 7: Do the authors have data on the Th1/Th2 balance in macaques induced with this regimen?

Response 7:

Thank you for your question regarding the Th1/Th2 balance in macaques induced with this regimen. While we did not directly assess Th2 cytokines (e.g., IL-4, IL-5) in this study, our ICS data revealed robust IFN-γ, IL-2 and TNF-α production by both CD4⁺ and CD8⁺ T cells (revised Figure 6), consistent with a Th1-polarized immune response. We acknowledge that characterizing the Th1/Th2 balance more comprehensively—for example, by quantifying IL-4, IL-5, or IL-13—would provide deeper mechanistic insights. Future studies will incorporate these analyses to evaluate the full cytokine polarization profile induced by heterologous regimens.

  • Comments 8: Cite the reference for the sentence in lines 382-384 in Discussion.

Response 8:

Thanks for your suggestion. We have added the relevant reference.

Reviewer 3 Report

Comments and Suggestions for Authors

The authors analyzed T cell responses in non-human primates that were vaccinated with a heterologous 4 vector  regimen against HIV-1 gag over almost 9 years. Before each new vaccination T cell responses dropped to levels below detection. The periods in which T cell responses were detectable tended to become longer after each additional vaccination. The study is carefully designed and clearly presented. The long 10-year follow-up is a strength. I have a few suggestion for improving the manuscript.

Comparisons between the 2 and 4 vector group should be made with caution, as only one animal received the 2 vector regimen and the variation within the 4 vector group is high. This should be pointed out more clearly, when conclusion on these differences are presented. 

Please add the individual ages of the macaques at first vaccination. 

The first part of the discussion is a second introduction and should be deleted. 

Fig. 4. The labeling of the x axis is too small.

Fig 6. Please describe in the legend what the horizontal dotted lines stand for. 

Fig 6b. It  seems that the binding antibodies to the Ad5 vector were already boosted before the 2nd dose of the Ad5-vaccine was applied. Is the labeling of the axis and the arrow for vaccine application correct?

Line 360: Why do you believe that lack of variation of neutralizing antibodies to Ad5 shows that vector neutralization did not reduce the efficacy of the subsequent Ad5 vaccination?

The 4 vector group  clearly had neutralizing antibodies to Ad5 before the final Ad5 vaccination, most likely from the first Ad5 double vaccination at week 6. However, it surprises that the animal in the 2 vector group did not develop Ad5 neutralizing antibodies after 2 doses of Ad5. Was there any difference between the early and the late Ad5 vaccinations (vector production/purification, method of quantification, vector formulation)? Was the 2 vector animal older than the 4 vector animals at the first Ad5 vaccination?

Conclusions: Line 477. I would use prolonged and not durable, as the T cells responses dropped after the final vaccination again after a year. 

Reference 10: Hear the original article and not the correction should be cited. 

Comments on the Quality of English Language

Careful review of  english grammar and spelling should be performed.

I am not a native speaker, and did not specifically look for grammar or spelling mistakes,  but  I detected a few potential errors: 

Line 33: In the absence of a cure, developing safe and effective vaccines remains is crucial for controlling the ...

Line 66: This extend investigation addresses critical unanswered questions: How durable are the...

Line 269: This intervention triggered a rapid surged in immune responses, peaking...

Line 435: While any vector proven safe and effectively in delivering 
foreign genes could sever as a vaccine platform, im.....

Author Response

The authors are deeply grateful to the reviewers for their insightful comments, which have been instrumental in enhancing the quality of this manuscript. After thorough analysis of all the issues raised and careful consideration of the suggestions, the authors have made appropriate responses and revisions. Below is a point - by - point reply to the reviewers' comments.

(Line numbers are based on the clean_revised version (PDF).)

Reviewer 3

Comments and Suggestions for Authors

The authors analyzed T cell responses in non-human primates that were vaccinated with a heterologous 4 vector  regimen against HIV-1 gag over almost 9 years. Before each new vaccination T cell responses dropped to levels below detection. The periods in which T cell responses were detectable tended to become longer after each additional vaccination. The study is carefully designed and clearly presented. The long 10-year follow-up is a strength. I have a few suggestion for improving the manuscript.

  • Comments 1: Comparisons between the 2 and 4 vector group should be made with caution, as only one animal received the 2 vector regimen and the variation within the 4 vector group is high. This should be pointed out more clearly, when conclusion on these differences are presented.

Response 1:

We fully acknowledge this limitation. The small sample size in the two-vector group (n=1) was due to a mistake at the beginning of this four-vector vaccine experiment. The animal 98R00237 was originally in the control group. It was vaccinated with rMVA vaccine by mistake at weeks 206. It could only be separated from the control group as a separate group. In the revised Result (Section 3.2) and Discussion (Section 4), we have added the following statement:

Section 3.2: The four-vector regimen demonstrated substantially greater magnitude and duration of immune responses compared to the single animal in the two-vector cohort (revised Figure 4). While formal statistical comparison was precluded by the two-vector group's sample size (n=1), qualitative differences were evident: the four-vector group maintained responses >500 SFCs/10⁶ PBMCs for 69 weeks versus 12 weeks in the two-vector macaque. (line 293-295)

Section 4: The limited sample size in the two-vector group (n=1) precludes definitive comparative conclusions, though observed response patterns align with established principles of heterologous prime-boost immunology. (line 503-506)

  • Comments 2: Please add the individual ages of the macaques at first vaccination.

Response 2:

The individual ages of the macaques at the start of the study have been added to the Materials and Methods section (Section 2.3):

“Macaque IDs and ages at first vaccination: 98R0053 (7 years), 00R0019 (5 years), 98R0013 (7 years), 00R0227 (5 years), 99R0035 (6 years), 98R0037 (7 years), 98R0009 (7 years), 00R0023 (5 years).” (line 128-130)

  • Comments 3: The first part of the discussion is a second introduction and should be deleted.

Response 3:

We have revised the discussion section to remove redundant introductory content. The revised discussion now directly addresses the study’s findings and their implications (section 4).

  • Comments 4: 4. The labeling of the x axis is too small.

Response 4:

The x-axis labels in Figure 4 have been enlarged for clarity (revised Figure 5).

  • Comments 5: Fig 6. Please describe in the legend what the horizontal dotted lines stand for.

Response 5:

The horizontal dotted lines were temporary lines added during analysis with no actual meaning. In the revised Figure 7, we've removed them to prevent confusion.

  • Comments 6: Fig 6b. It  seems that the binding antibodies to the Ad5 vector were already boosted before the 2nd dose of the Ad5-vaccine was applied. Is the labeling of the axis and the arrow for vaccine application correct?

Response 6:

We confirm that the timeline labels and vaccination arrows in revised Figure 7b are accurate. The apparent increase in Ad5-binding antibody titers at week 326 (prior to the second rAd5 vaccination at week 336) reflects assay variability or individual animal variation. Specifically:

In the 4-vector group, only one animal showed a modest 2-fold increase (100 to 200), while the remaining three animals exhibited stable titers.

In the 2-vector group, a similar 2-fold increase (100 to 200) was observed in the single animal.

Notably, after the second rAd5 vaccination at week 336, titers rose markedly by week 338:

4-vector group: Three of four animals showed a 4-fold increase (200 to 800).

2-vector group: The single animal’s titer increased from 200 to 400.

These data suggest that pre-existing Ad5-binding antibodies did not significantly blunt the booster effect, and the observed variability likely stems from biological or technical fluctuations.

We have added this clarification to the Results section (line 403-405).

  • Comments 7: Line 360: Why do you believe that lack of variation of neutralizing antibodies to Ad5 shows that vector neutralization did not reduce the efficacy of the subsequent Ad5 vaccination?

Response 7:

Thanks for your question.

The stable Ad5-neutralizing antibody (NAb) titers following the second Ad5 vaccination indicate that pre-existing vector immunity remained consistent, suggesting the primary Ad5 immunization effectively primed the humoral response without eliciting further NAb amplification upon boosting. This observation implies minimal interference of vector-specific immunity with subsequent antigen delivery. Notably, while Ad5-binding antibodies increased post-second vaccination (week 336), reflecting active immune recognition of the vector, this did not correlate with a proportional rise in NAbs that could compromise vaccine efficacy. Critically, despite stable NAb levels, the heterologous regimen maintained robust cellular immunogenicity, as evidenced by a secondary ELISPOT response peak at week 338 (two weeks post-second rAd5 dose) and sustained IFN-γ⁺ T-cell responses exceeding 500 SFCs/10⁶ PBMCs for 69 weeks (weeks 336–405, Figure 4). These findings collectively demonstrate that the sequential heterologous strategy successfully balances the induction of durable cellular immunity with controlled vector-specific humoral responses. (line 413-426)

  • Comments 8: The 4 vector group  clearly had neutralizing antibodies to Ad5 before the final Ad5 vaccination, most likely from the first Ad5 double vaccination at week 6. However, it surprises that the animal in the 2 vector group did not develop Ad5 neutralizing antibodies after 2 doses of Ad5. Was there any difference between the early and the late Ad5 vaccinations (vector production/purification, method of quantification, vector formulation)? Was the 2 vector animal older than the 4 vector animals at the first Ad5 vaccination?

Response 8:

We appreciate the reviewer’s insightful question。

All Ad5 vaccines used in this study were produced under identical protocols (purification, formulation, and quantification methods), as described in Section 2.1. While the early (week 6) and late (week 336) Ad5 vaccinations were from different production batches, rigorous quality control ensured consistency in viral particle (VP) titers and immunogenicity.

We acknowledge that the age difference between groups likely influenced humoral responses. The 4-vector animals were aged 5–7 years at their first Ad5 vaccination (week 6), while the 2-vector animal was ~12.5 years old at its first Ad5 vaccination (week 282). Advanced age is associated with immunosenescence, which may dampen antibody production. Despite this, robust cellular immunity was preserved in both groups (revised Figure 4). The small sample size (n=1 in the 2-vector group) further limits definitive conclusions.

  • Comments 9: Conclusions: Line 477. I would use prolonged and not durable, as the T cells responses dropped after the final vaccination again after a year.

Response 9:

We have revised the conclusion to state:

“induce potent and prolonged” (Line 517).

  • Comments 10: Reference 10: Hear the original article and not the correction should be cited.

Response 10:

Reference 10 has been corrected to cite the original article:

Xiao, M.; Xie, L.; Cao, G.; Lei, S.; Wang, P.; Wei, Z.; Luo, Y.; Fang, J.; Yang, X.; Huang, Q., et al. CD4(+) T-cell epitope-based heterologous prime-boost vaccination potentiates  anti-tumor immunity and PD-1/PD-L1 immunotherapy. J Immunother Cancer 2022, 10, doi:10.1136/jitc-2021-004022. (line 606-608)

  • Comments 11: Comments on the Quality of English Language

Careful review of  english grammar and spelling should be performed.

I am not a native speaker, and did not specifically look for grammar or spelling mistakes,  but  I detected a few potential errors:

Line 33: In the absence of a cure, developing safe and effective vaccines remains is crucial for controlling the ...

Line 66: This extend investigation addresses critical unanswered questions: How durable are the...

Line 269: This intervention triggered a rapid surged in immune responses, peaking...

Line 435: While any vector proven safe and effectively in delivering foreign genes could sever as a vaccine platform, im.....

Response 11:

Line 33: “remains is crucial” → “remains crucial”. (line 34)

Line 66: “extend investigation” → “extended investigation”. (Paragraph revision)

Line 269: “surged” → “surge”. (line 318)

Line 435: “effectively” → “effective”; “sever” → “serve”. (line 482-483)

Thank you for your helpful suggestions. We have carefully checked the grammar and spelling of the paper. We will also introduce some changes to the text to make the paper clearer to readers.

Round 2

Reviewer 1 Report

Comments and Suggestions for Authors

The authors have revised the article in line with the recommendations, so I have no further questions for them.